# Early Growth Characterization and Antioxidant Responses of *Phellodendron chinense* Seedling in Response to Four Soil Types at Three Growth Stages

Yao Yang [1], Yun-Yi Hu [1], Wen-Zhang Qian [1], Ya-Juan Wang [1], Hong-Yu Ren [1], Shun Gao [1,2] and Guo-Xing Cao [1,2,*]

1 Department of Forestry, Faculty of Forestry, Sichuan Agricultural University, Chengdu 611130, China; yangyao@stu.sicau.edu.cn (Y.Y.); huyunyi@stu.sicau.edu.cn (Y.-Y.H.); 202001247@stu.sicau.edu.cn (W.-Z.Q.); wangyajuan@stu.sicau.edu.cn (Y.-J.W.); renhongyu@stu.sicau.edu.cn (H.-Y.R.); shungao@sicau.edu.cn (S.G.)
2 National Forestry and Grassland Administration Key Laboratory of Forest Resources Conservation and Ecological Safety on the Upper Reaches of the Yangtze River & Sichuan Province Key Laboratory of Ecological Forestry Engineering on the Upper Reaches of the Yangtze River, Sichuan Agricultural University, Chengdu 611130, China
* Correspondence: cgxing@sicau.edu.cn

**Abstract:** Soil type is an important environmental factor that affects plant growth and development, especially during the early growth stage. In this study, *P. chinense* (*Phellodendron chinense*) seedlings were cultivated on red soil (RS), yellow soil (YS), acidic purple soil (ACPS), and alkaline purple soil (ALPS), and the influence of soil types on the growth parameters and physiological responses at three growth stages were investigated. The results show that soil types and growth stages can significantly influence seedling height, base diameter, and biomass in *P. chinense* seedlings. Moreover, the significant variations in chlorophyll, total polyphenols, soluble protein, soluble sugar, and malondialdehyde (MDA) content, as well as superoxide dismutase (SOD), peroxidase (POD), phenylalanine ammonia-lyase (PAL), and polyphenol oxidase (PPO) activity, were recorded at three growth stages among four soil types. These results show that *P. chinense* seedlings can grow normally in four soil types, and ACPS may be more beneficial for the artificial cultivation of *P. chinense* seedlings than those of RS, ALPS, and YS. Principal component analysis (PCA) reveals a strong correlation and clear differences in the test parameters between growth stage and organs in four soil types, and the cumulative contribution percentages of the total biplot range from 74.44% to 81.97%. This present information will be helpful for farmers in selecting soil type for the large-scale cultivation of *P. chinense* seedlings.

**Keywords:** antioxidant enzyme; growth parameters; physiological traits; *P. chinense*; principal component analysis; soil properties





## 1. Introduction

The seedling stage is the most critical life stage for the normal development of plant systems and processes, while the demands of different seedlings have different demands for suitable soil conditions due to different needs of nutrients, water, suitable pH, etc. [1]. During the early seedling formation process, the growth characteristics and physiological and biochemical parameters of seedlings are strongly affected by biotic and abiotic factors, such as shade, drought, waterlogging, soil type, etc. Among these, soil types represent variable physiochemical and microbiological properties, such as soil texture, water-holding capacity, aggregate size, porosity, pH, organic matter, and mineral nutrients, etc. These parameters are known to affect plant growth and development, productivity, and biochemical compositions [2,3]. Kahkashan et al. found that plant growth parameters in different soil types were significantly different, and plants growing in sandy clay had the fastest

growth and the highest content of essential oil yield in *Mentha arvensis* [2]. When comparing *Artemisia annua* plants under clay loamy and sand loamy soils, higher vegetative growth characteristics were obtained under clay loamy soil [4]. Compared to those of yellow soil and red soil, the biomass of *Firmiana simplex* seedlings reaches its maximum value when cultivated in acidic purple soil [5]. Some studies have shown that higher soil moisture and nutrient availability are more important in the early stages than the later stages of seedling growth, probably because light becomes the main limiting resource in older forests [6,7]. Sinclair also reports that the growth performance of plants is positively related to the available minerals in the soil [8]. These reports indicate that soil types are directly related to growth, and the development of physiological and biochemical parameters, as well as seedling quality of plants. Moreover, inter- and intra-competition with seedlings and grasses also play a major role in determining the seedlings growth [9]. Thus, it is necessary to study the changes and modifications caused by differences in soil types and screen suitable soil types to obtain high-quality seedlings of specific species.

Being an abiotic factor, soil types may induce oxidative stress during the early seedling formation process, which may be linked to the excess production of reactive oxygen species (ROS) [10]. In order to balance the excess ROS, plants may induce non-enzymatic (soluble sugars, soluble protein, polyphenol, and amino acid, etc.) and enzymatic (superoxide dismutase (SOD), catalase (CAT), peroxidase (POD), phenylalanine ammonialyase (PAL), polyphenol oxidase (PPO)) systems, enabling them to counteract these oxidative damages [11]. Early reports show that the activities of various antioxidant enzymes and non-enzymatic components in plants generally increase after exposure to various environmental stresses [12]. Previous studies mentioned that different soils (salinity, water content, element content, etc.) affected antioxidant enzyme activities in plants [13–15], and was directly related to seed germination and early seedling survival [16,17]. Thus, changes in non-enzymatic and enzyme components are affected by competition or facilitation under different types of soil, which may be considered as typical defensive components during early seedling formation.

*Phellodendron chinense* (*P. chinense*), belonging to the genus *Phellodendron* (Rutaceae), known as "Chuan Huang Bo" in China, is one of the traditional Chinese woody medicinal plants. Its stem and root bark are the commonly used parts, and are rich in more than 10 types of effective components, including alkaloids, etc., which are mainly used to treat several important diseases with a long medicinal history [18]. Recently, *P. chinense* and its products have attracted the interest of researchers for meeting the increasing requirements of market demand. Large-scale plantations of *P. chinense* play an increasingly important role in solving national and international market demand. *P. chinense* is mainly distributed in various soil types in Southwest China, and Sichuan has the largest cultivation area. There is a wide range of *P. chinense* plantations in Southwest China, which respond differently under various soil types. Although *P. chinense* can grow normally in a variety of soil types, of which chemical ingredients vary significantly from different geographical locations, obtaining the maximum yield is dependent on knowledge of the best-suited soil types. The main reason is that the growth parameters of medicinal plants and their internal chemical composition content are closely related to their surrounding environment, where they come in contact with different abiotic components such as water, light, temperature, soil, and chemicals, forming different soil type and climatic conditions [19]. Moreover, the harvest period of bark is generally 8–12 years. The quality of the seedling is basic for large-scale cultivation of *P. chinense* plantation and directly affects the growth cycle, the content of active components, and the potential yield of *P. chinense* bark. Thus, propagating healthy seedlings plays a critical role in plantation establishment. Several reports are available on the effects of different regions on chemical constituents in the leaves, fruit, and bark of *P. chinense* [18,20]. Our previous studies showed that N may effectively increase seedling height and ground diameter, and affected biomass accumulation and distribution, photosynthetic characteristics in *P. chinense* seedlings cultured in red soil, alkaline purple soil, and acidic purple soil, and it grew better in acidic purple soil [21]. However, most of

the studies were carried out in the late seedling stage, and there are still largely unknown growth parameters and physiological characteristics during early seedling growth stages of most medicinal plants under different soil types. In this study, the effects of red soil (RS), yellow soil (YS), alkaline purple soil (ALPS), and acidic purple soil (ACPS) on growth, physiological, and biochemical changes of *P. chinense* seedlings at early growth stages were analyzed. The objective of this research was to understand the independent and interactive effects of soil type and growth stages on growth parameters and antioxidant defensive systems, which will provide practical information for adjusting management practices at early seedling growth stages and optimizing seedling cultivation strategy.

## 2. Materials and Methods

### 2.1. Study Site

The study site was located in a greenhouse at Sichuan Agricultural University, Chengdu, China at latitude 30°38′ N, longitude 103°45′ E, and altitude of 505 m. This site belongs to a mid-latitude inland subtropical humid monsoon climate with an annual average temperature of 16.4 °C, and average annual precipitation of 1814 mm, mostly falling in summer with an average relative air humidity of 84%. The annual frost-free period is 282 days, and the total annual sunshine is 1104.5 h.

### 2.2. Soil Sampling and Chemical Analysis

The RS, YS, ALPS, and ACPS were collected from Sanxing Village, Fengle Township, Shimian County, Yaan City, China (N29°32′, E102°54′, H878 m), Baisheng Village, Baolin Town, Qiong-lai, Chengdu, China (N30°21′, E103°30′, H552 m), Jifeng Town, Zhong-jiang County, De-yang, China (N31°03′, E104°68′, H900 m), and Boshan Reading Park, Yucheng District, Ya'an City, Sichuan Province (N29°58′, E102°58′, H679 m). The soil was collected by removing 5 cm of topsoil, then digging down 30 cm and packing the excavated soil back into the laboratory. The four soils were sieved using a 4 mm sieve to remove roots and rocks and potted for use, and the basic physical and chemical properties of the four soils were determined (Table 1). Soil pH was determined using an electrode pH meter in a soil:water (1:2) suspension. Total nitrogen content was determined by the Kjeldahl method. The soil moisture and organic matter content were determined by loss on ignition method [22].

### 2.3. Seedling Culture and Growth Parameters Measurement

The experiment was laid out as a single factor completely randomized design with nine replications for each soil type from March 2019 to August 2019. One hundred and twenty pots were prepared for four types of soil, thirty each of RS, YS, ALPS, and ACPS. Healthy seeds were treated and germinated using the following steps. In brief, *P. chinense* seeds were harvested from Xie-yuan Town, Da-yi County, Chengdu, China. In brief, seeds were sterilized with 0.1% potassium permanganate solution for 30 min, and then washed with water. These seeds were incubated at 40 °C for 24 h and 10 seeds were sowed in each plastic pot (diameter and height, 37 × 27 cm) with 5 ± 0.5 kg of four types of soil. The pots were placed in a greenhouse. Weak seedlings were removed 7 days after germination until the cotyledons were fully expanded, eventually leaving only one seedling per pot to prevent competition between seedlings from affecting the results of the experiment. Irrigation, weeding, and disease prevention management were carried out two to three times weekly according to the observed needs. After 30, 60, and 90 days, seedling height was measured using a tape ruler. Stem basal diameter was measured by a vernier caliper at the base of the stem. At each sampling time, ten seedlings were randomly harvested and sectioned into roots, stems, and leaves. All plant parts were cleaned with deionized water, and the fresh weight (FW) was measured. Then, these samples were dried at 105 °C for 30 min in an oven and finally, plant organs were dried to a constant weight under 70 °C for 48 h. Then, the different organs of each seedling were weighed to determine the dry

weight (DW) of the leaf, stem, and root. Moreover, fresh samples were stored at $-80$ °C for protein extraction and enzyme analyses.

**Table 1.** Background values of soil physicochemical properties.

| No. | Yellow Soil | Red Soil | Acidic Purple Soil | Alkaline Purple Soil |
|---|---|---|---|---|
| pH | 4.805 ± 0.018 b | 4.990 ± 0.075 b | 4.343 ± 0.044 c | 8.733 ± 0.105 a |
| Moisture content (%) | 0.115 ± 0.012 a | 0.021 ± 0.006 c | 0.120 ± 0.002 a | 0.05 ± 0.001 b |
| Total nitrogen (g·kg$^{-1}$) | 17.492 ± 1.580 b | 11.301 ± 0.431 c | 28.513 ± 0.163 a | 15.906 ± 0.884 b |
| Ammonium nitrogen (mg·kg$^{-1}$) | 19.784 ± 0.916 a | 18.667 ± 0.362 a | 11.123 ± 1.214 b | 13.259 ± 1.971 b |
| Nitrate nitrogen (mg·kg$^{-1}$) | 55.696 ± 2.207 a | 7.190 ± 0.563 c | 26.410 ± 0.895 b | 2.051 ± 0.571 d |
| Total phosphorous (g·kg$^{-1}$) | 0.076 ± 0.005 c | 0.069 ± 0.006 c | 0.235 ± 0.007 b | 0.301 ± 0.011 a |
| Available phosphorous (g·kg$^{-1}$) | 0.0546 ± 0.004 b | 0.0338 ± 0.001 c | 0.0512 ± 0.002 b | 0.065 ± 0.002 a |
| Total potassium (g·kg$^{-1}$) | 13.460 ± 0.251 b | 8.844 ± 0.775 c | 12.170 ± 0.200 b | 19.180 ± 0.851 a |
| Organic matter (g·kg$^{-1}$) | 42.014 ± 8.623 a | 16.643 ± 0.445 b | 42.527 ± 2.655 a | 20.989 ± 0.425 b |

Note: lowercase letters represent significant differences in different soil types, with a significance level of 0.05.

### 2.4. Determination of Chlorophyll Contents

The chlorophyll content was determined using the acetone method [23]. The fresh leaves of 100 mg were placed into a stopper tube containing 10 mL of an acetone–ethanol mixture (1:1, *v/v*), and then stewed in the dark until the leaves were completely white. The absorbance was measured at 663 nm and 645 nm with the mixture of acetone and ethanol as the reference solution, and the chlorophyll content was calculated.

### 2.5. Estimation of Soluble Protein (SP) and Total Polyphenols (TP) Contents

Fresh samples (0.2 g) were ground with liquid nitrogen and homogenized in 50 mM sodium phosphate buffer (2 mL, pH 7.0) containing 0.5 mM EDTA and 0.15 M NaCl. The supernatant was harvested by centrifuging at 12,000 rpm for 10 min at 4 °C and used for SP content, and assaying of SOD, POD, and PPO activity. SP content was determined by Coomassie brilliant blue G-250 method using (BSA) as the standard [24]. Fresh samples (0.5 g) were ground with liquid nitrogen, and extracted with 10 mL of 80% ethanol at 100 °C for 30 min, and the supernatant was harvested by centrifuging at 10,000 rpm at 4 °C. The reaction solution contained 0.4 mL extraction and 0.4 mL of Folin–Ciocalteau phenol reagent, 1.2 mL 7.5% $Na_2CO_3$ solution, and 2 mL distilled water, and incubated at room temperature for 2 h. The absorbance was measured at 760 nm [25], and results were expressed in mg of gallic acid/g of dry weight.

### 2.6. Determination of Malondialdehyde (MDA) and Soluble Sugar (SS) Contents

MDA and SS contents were estimated using the Heath and Packer method [26]. Leaves, stems, and roots (0.2 g) were ground with liquid nitrogen and homogenized in 5 mL of 10% trichloroacetic acid (TCA). The supernatant was harvested by centrifuging at 12,000 rpm for 5 min at 4 °C. The reaction mixture included 1.0 mL of the supernatant, and 4.0 mL of 0.5% TBA in 5% TCA, and was heated at 100 °C for 30 min and then cooled in an ice bath. The samples were centrifugated at 12,000 rpm for 10 min, and the absorbance was recorded at 450 nm, 532 nm, and 600 nm. MDA content was calculated according to the following formula: MDA concentration (μM) = $6.45(A_{532} - A_{600}) - 0.56A_{450}$, and was expressed as μM per gram fresh weight. SS contents were calculated according to the following formula: SS content (μM) = $11.71A_{450}$, and was expressed as μM per gram fresh weight.

### 2.7. Determination of Superoxide Dismutase (SOD), Peroxidase (POD), Phenylalanine Ammonialyase (PAL), and Polyphenol Oxidase (PPO) Activities

The activities of SOD and POD were assayed by the nitro-blue tetrazolium (NBT) reduction method and guaiacol method, respectively [27]. The 50% inhibition of NBT actinic reduction was taken as a SOD activity unit (U). One unit of POD activity was defined as the amount of enzyme that produces 1.0 absorbance change at 470 nm per min.

PPO activity was assayed by the method of He et al. [28]. One unit of PPO activity was defined as the amount of enzyme causing 1 absorbance increase per min. As for PAL activity, fresh samples were ground in ice-cold Tris-HCl buffer (50 mM, pH 8.8) containing 1% polyvinylpolypyrrolidone and 0.1 mM EDTA. The supernatant was harvested by centrifuging at 12,000 rpm for 5 min at 4 °C, and PAL activity was determined by monitoring the reaction product trans-cinnamate at 290 nm [27]. One unit of PAL activity was defined as the amount of enzyme that increased the absorbance by 0.01 per min. Data were expressed as enzyme units per gram fresh weight (U/g FW).

### 2.8. Statistical Analysis

Experiments were carried out in a randomized way with three replicates. Data were analyzed using one-way analysis of variance (one-way ANOVA) and expressed as means ± S.E. Duncan's multiple comparison tests at $p \leq 0.05$, regression analysis, and correlation analysis were performed for evaluating the significance differences among different groups using SPSS 20.0 (SPSS Inc., Chicago, IL, USA). To explore and visualize the relationships between tested parameters and treatments, principal component analysis (PCA) was performed using OriginPro 2021 (OriginLab Corporation, Northampton, MA, USA). For PCA, the variables were transformed into a dimensionless standardized form, and the standardized data were used to obtain the major contributing factors via eigenvector decomposition. The results of all PCA analyses were assessed only for factors with eigenvalues > 1.

## 3. Results

### 3.1. Growth Parameter and Chlorophyll Contents

As shown in Figure 1A–C, the biomass of roots and stems at three growth stages show significant differences when *P. chinense* seedlings are cultured in four soil types, whereas the leaves are only significantly different in the early stage. The biomass of leaves, stems, and roots reach the maximum in red soil at 30 days of culture, and the values represent 0.673, 0.214, and 0.245 g, respectively. However, the lowest biomass in the leaves, stems, and roots are observed when cultured in ACPS, and the values represent 0.359, 0.135, and 0.108 g, respectively. Moreover, *P. chinense* seedlings cultured in RS are significantly superior to those grown in ACPS, for example at 30 days. However, as the seedling's growth continued up to 90 days, the biomass of different organs cultured in ACPS was remarkably higher than those of in RS, YS, and ALPS (Figure 1B,C). As shown in Figure 1D, the chlorophyll content cultured in RS, ACPS, and ALPS gradually decreases with the rising culture time up to 90 days, and the maximum values of 1.87, 1.76, and 1.56 mg/g, respectively, are observed at day 30. However, when seedlings are cultured in YS, the highest values of 2.62 mg/g are found at day 60. As shown in Figure 1E,F, the significant variations in plant height and basal diameter of *P. chinense* seedlings are also observed when cultured in four soil types at three growth stages. At the later stage (90 days), the plant height and basal diameter reach the maximum of 23.79 cm and 0.559 cm, respectively, when seedlings are cultured in ACPS. These results suggest that the changed growth parameters of different organs and chlorophyll content in *P. chinense* seedlings are related to soil types and growth stages.

### 3.2. Soluble Sugar (SS) and Soluble Protein (SP) Contents

As shown in Figure 2A–C, SS content in leaves, stems, and roots shows larger differences among soil types at three growth stages. At 30 days, the SS content ranges from 186.6 to 252.0 μmol/g in the leaves, from 90.5 to 129.0 μmol/g in the stems, and from 52.9 to 145.7 μmol/g in the roots. The maximum values in the leaves, stems, and roots are observed in ACPS, ALPS, and YS, respectively. At day 60 and 90, the SS content in the different organs is significantly less than those at 30 days and shows remarkable variations in the four soil types. These results show that the SS content in different organs of *P. chinense* seedlings is associated with soil types and growth stages. As shown in Figure 2D–F, the SP content in different organs of *P. chinense* seedlings exhibits significant differences at

three growth stages and in four soil types. At day 30, the SP content ranges from 26.6 to 43.1 mg/g in the leaves, from 11.8 to 13.7 mg/g in the stems, and from 5.24 to 7.56 mg/g in the roots. The maximum values in the leaves, stems, and roots are observed in RS, ALPS, and ALPS, respectively. At day 60, the SP content in different organs is significantly lower than those of day 30. Similarly, at day 90, the SP content in the leaves is significantly lower than those of at day 60. However, the SP content in the stems and roots show an opposite trend. These results show that the SP content of different organs of *P. chinense* seedlings has different responses to growth stages and four soil types.

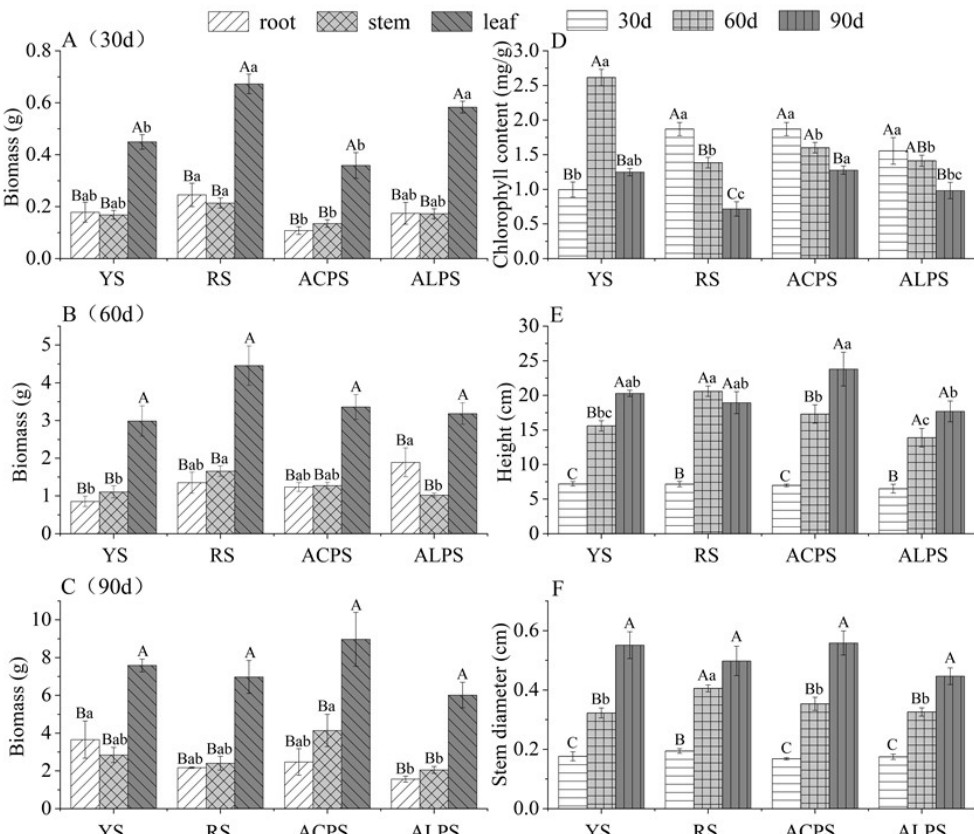

**Figure 1.** Effects of soil types on growth parameters and leaf chlorophyll contents in *P. chinense* seedlings at three growth stages. (**A**), Biomass of leaves, stems and roots of seedlings in four soil types at 30 day. (**B**), Biomass of leaves, stems and roots of seedlings in four soil types at 60 day. (**C**), Biomass of leaves, stems and roots of seedlings in four soil types at 90 day. (**D**), Chlorophyll content in leaves at three growth stages in four soil types. (**E**), Plant height of seedlings at three growth stages in four soil types. (**F**), Stem diameter of seedlings at three growth stages in four soil types. YS, yellow soil. RS, red soil. ACPS, acidic purple soil. ALPS, alkaline purple soil. Data represent mean ± S.E., *n* = 3. Lowercase letters represent significant differences in different soil types, and capital letters represent significant differences in different organs, with a significance level of 0.05.

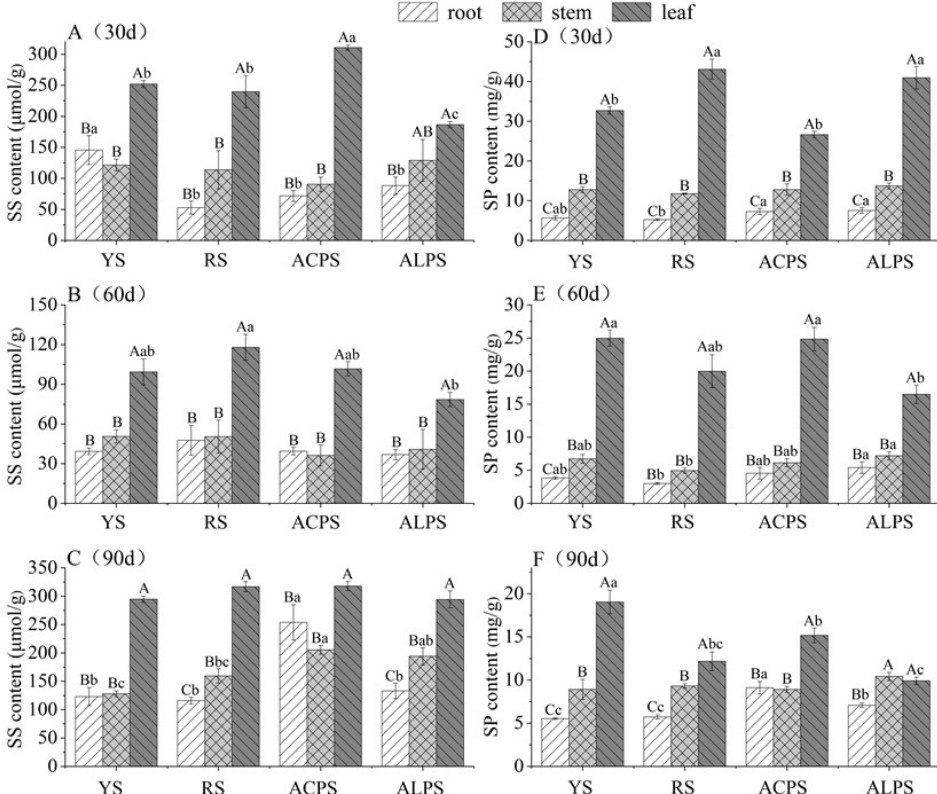

**Figure 2.** Effects of soil types on soluble sugar (SS) and soluble protein (SP) content in *P. chinense* seedlings at three growth stages. (**A**), SS content in leaves, stems and roots of seedlings at 30 days in four soil types. (**B**), SS content in leaves, stems and roots of seedlings at 60 days in four soil types. (**C**), SS content in leaves, stems and roots of seedlings at 90 days in four soil types. (**D**), SP content in leaves, stems and roots of seedlings at 30 days in four soil types. (**E**), SP content in leaves, stems and roots of seedlings at 60 days in four soil types. (**F**), SP content in leaves, stems and roots of seedlings at 90 days in four soil types. YS, yellow soil. RS, red soil. ACPS, acidic purple soil. ALPS, alkaline purple soil. Data represent mean ± S.E., *n* = 3. Lowercase letters represent significant differences in different soil types, and capital letters represent significant differences in different organs, with a significance level of 0.05.

### 3.3. Malondialdehyde (MDA) and Total Polyphenol (TP) Contents

As shown in Figure 3A–C, the results show that there are higher variations in MDA content in different organs of *P. chinense* seedlings than those at different stages and among four soil types. At day 30, the MDA content in the leaves varies from $2.19 \times 10^{-2}$ to $4.07 \times 10^{-2}$ mmol/g, ranges from $1.15 \times 10^{-2}$ to $1.44 \times 10^{-2}$ mmol/g in the stems, while the content in the roots ranges from $0.90 \times 10^{-2}$ to $1.98 \times 10^{-2}$ mmol/g. The MDA levels of leaves, stems, and roots reach their maximum values in ACPS, ALPS, and YS, respectively. When cultured for 60 days, the MDA content in the leaves, stems, and roots reach their maximum values in RS, these values being 5.45 mmol/g, 3.54 mmol/g, and 3.55 mmol/g, respectively. With the rising culture time up to 90 days, the MDA content of different organs significantly increases in four soil types, and these values are significantly higher than those of 30 and 60 days. As shown in Figure 3D–F, there are significant differences in the TP content in various organs cultured in four soil types at three growth stages. At 30 days of cultivation, the TP content in the leaves, stems, and roots reaches its maximum value when seedlings are cultured in ACPS, YS, and ACPS, representing 6.80 mg/g, 1.96 mg/g, and 0.69 mg/g, respectively. When cultured for 60 days, the TP content in the leaves, stems, and roots reaches its maximum in ALPS, these values being 5.65 mg/g, 2.13 mg/g, and 0.933 mg/g, respectively. At 90 days of cultivation, the TP content in the leaves and stems reaches its maximum value in RS, the values being 9.51 mg/g and 1.97 mg/g, respectively.

However, the highest levels in the roots are found when seedlings are cultured in ACPS, and the peak levels are 0.765 mg/g. These results show that the variations in MDA and TP content in the different organs of *P. chinense* seedlings depend on soil types and growth stages. Moreover, it could be observed that the MDA and TP content in the leaves is significantly higher than those of in the stems and roots.

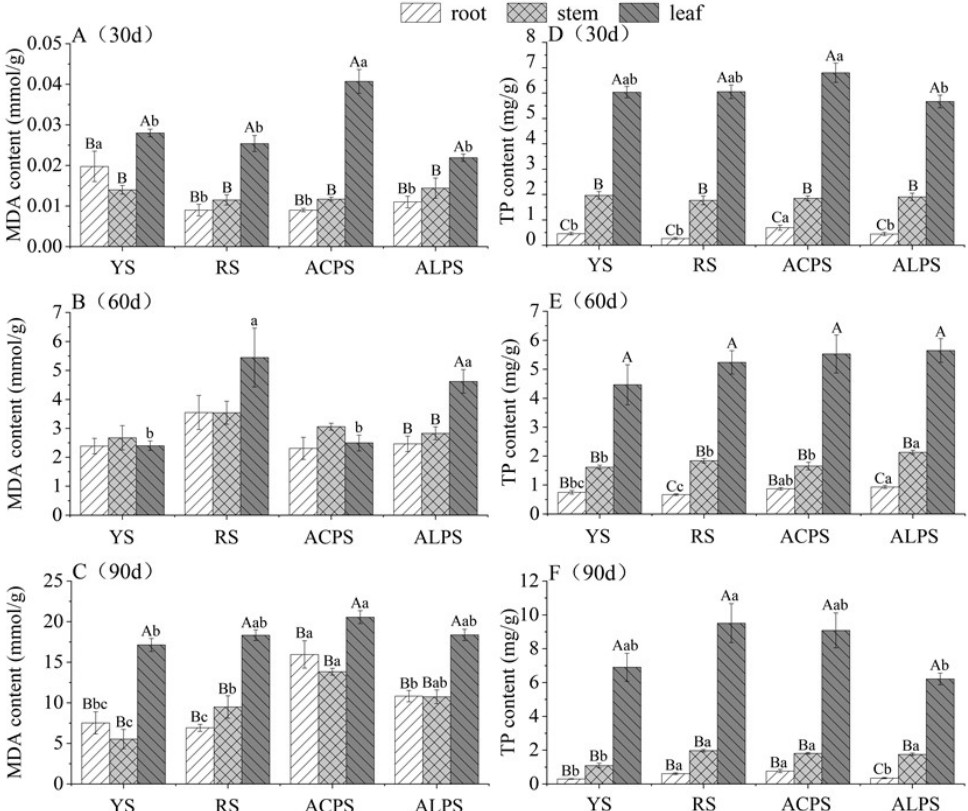

**Figure 3.** Effects of soil types on malondialdehyde (MDA) and total polyphenol (TP) content in *P. chinense* seedlings at three growth stages. (**A**), MDA content in leaves, stems and roots of seedlings at 30 days in four soil types. (**B**), MDA content in leaves, stems and roots of seedlings at 60 days in four soil types. (**C**), MDA content in leaves, stems and roots of seedlings at 90 days in four soil types. (**D**), TP content in leaves, stems and roots of seedlings at 30 days in four soil types. (**E**), TP content in leaves, stems and roots of seedlings at 60 days in four soil types. (**F**), TP content in leaves, stems and roots of seedlings at 90 days in four soil types. YS, yellow soil. RS, red soil. ACPS, acidic purple soil. ALPS, alkaline purple soil. Data represent mean $\pm$ S.E., $n = 3$. Lowercase letters represent significant differences in different soil types, and capital letters represent significant differences in different organs, with a significance level of 0.05.

## 3.4. Superoxide Dismutase (SOD) and Peroxidase (POD) Activities

As shown in Figure 4A–C, soil types, growth stages, and organs, as well as their interactions, have significant effects on SOD activity. At 30 days of incubation, the SOD activities in the leaf and stem show peak values in ACPS, the values being 715.5 U/g FW and 640.6 U/g FW, respectively. However, the highest SOD activities of 410.8 U/g FW in the roots are observed when seedlings are cultured in YS (Figure 4A). At 60 days of incubation, the maximum SOD activities of 322.4 U/g FW and 292.0 U/g FW in the leaves and roots are recorded in acidic purple soil, respectively, while the highest activity in the stems is found in alkaline purple soil (283.7 U/g FW) (Figure 4B). After 90 days of incubation, the highest SOD activities in the leaves, stems, and roots are observed when seedlings are cultured in the YS, RS, and ACPS, the values being 395.7 U/g FW, 230.1 U/g FW, and 244.2 U/g FW, respectively (Figure 4C). Moreover, the results show that SOD activities in the leaves are significantly higher than those in the stems and roots. As shown in Figure 4D–F, the POD activity clearly differs

in responses to soil types, growth stages, and organs in *P. chinense* seedlings. At 30 days of incubation, the POD activities in the stem and root reach the maximum values of 25.5 U/g FW and 17.0 U/g FW in ACPS, respectively, while the peak values for the leaves, with 10.0 U/g FW, are seen when seedlings are cultured in YS (Figure 4D). At 60 days of incubation, the POD activities in the leaves and stems reach maximum values with 2.74 U/g FW and 14.1 U/g FW, respectively, when seedlings are cultured in ACPS, while the maximum values in the roots are 12.9 U/g FW when cultured in YS (Figure 4E). At 90 d of incubation, the POD activities in the stems and roots reach maximal values with 22.8 U/g FW and 20.4 U/g FW, respectively, when seedlings are cultured in RS, while the activity in the leaves show peak values of 14.7 U/g FW in ACPS (Figure 4F). Moreover, the POD activities in the stems and roots are remarkably higher than those in the leaves in response to soil types and growth stages.

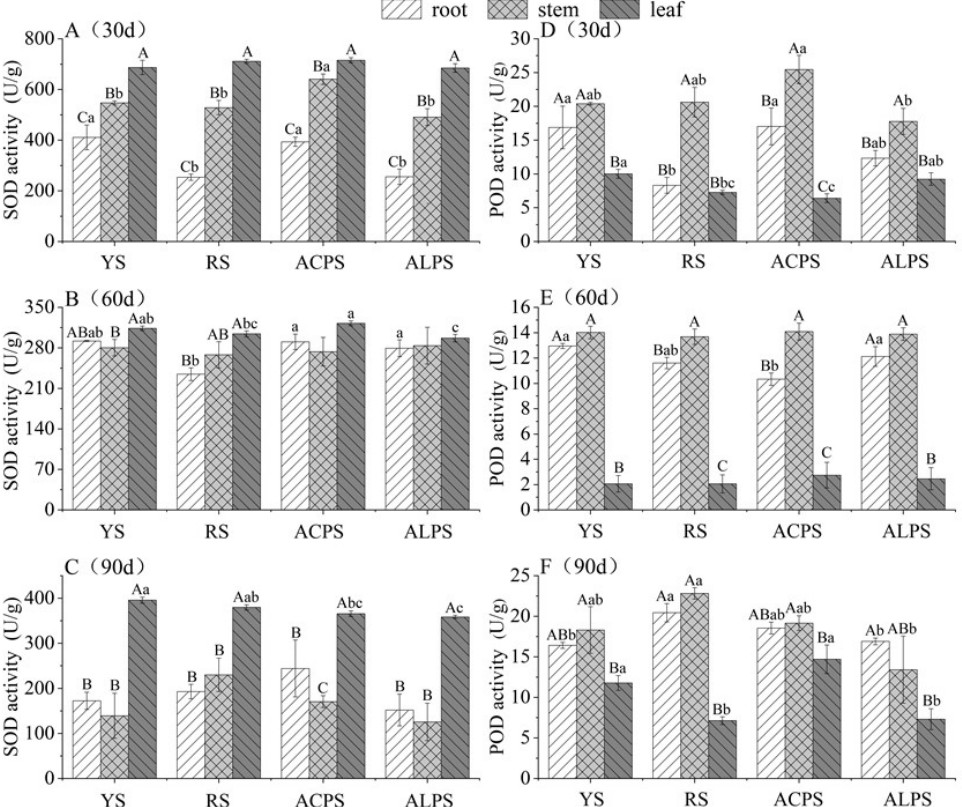

**Figure 4.** Effects of soil types on superoxide dismutase (SOD) and peroxidase (POD) activities in *P. chinense* seedlings at three growth stages. (**A**), SOD activities in leaves, stems and roots of seedlings at 30 days in four soil types. (**B**), SOD activities in leaves, stems and roots of seedlings at 60 days in four soil types. (**C**), SOD activities in leaves, stems and roots of seedlings at 90 days in four soil types. (**D**), POD activities in leaves, stems and roots of seedlings at 30 days in four soil types. (**E**), POD activities in leaves, stems and roots of seedlings at 60 days in four soil types. (**F**), POD activities in leaves, stems and roots of seedlings at 90 days in four soil types. YS, yellow soil. RS, red soil. ACPS, acidic purple soil. ALPS, alkaline purple soil. Data represent mean ± S.E., $n = 3$. Lowercase letters represent significant differences in different soil types, and capital letters represent significant differences in different organs, with a significance level of 0.05.

### 3.5. Phenylalanine Ammonia Lyase (PAL) and Polyphenol Oxidase (PPO) Activities

As shown in Figure 5A–C, soil types, growth stages, and organs, as well as their interactions, have significant effects on PAL activity. At day 30, the highest PAL activity in the leaves, stems, and roots is observed when seedlings are cultured in RS, ALPS, and YS, and the values represent 2.76 U/g, 2.56 U/g, and 1.46 U/g FW, respectively (Figure 5A). As shown in Figure 5B, at day 60, the leaves' PAL activity of 0.463 U/g is the highest when

seedlings are cultured in ACPS, and the PAL activity in the stems and roots cultured in YS (0.150 U/g) and ALPS (0.059 U/g) shows the highest values. As shown in Figure 5C, at day 90, the PAL activities in the leaves and stems reach maximal values when cultured in RS, at 0.816 U/g and 0.358 U/g FW, respectively, while PAL activities in the roots reach maximal values when cultured in ACPS (0.126 U/g FW). At 30 days of incubation, the highest PPO activity in the stems and roots is observed when cultured in ACPS, and the values represent 139.5 U/g and 107.3 U/g FW, respectively. However, the PPO activity in the leaves reaches the highest values when cultured in YS (Figure 5D). As shown in Figure 5E, at 60 days, the PPO activities in the stems and roots reach maximal values when cultured in ALPS, representing 37.7 U/g and 42.9 U/g FW, respectively, while the leaf PPO activities show maximal values when cultured in YS (87.6 U/g FW). As shown in Figure 5F, at 90 days, the PPO activities in the leaves, stems, and roots reach their maximum values when seedlings are cultured in YS (22.8 U/g), ALPS (11.5 U/g), and ACPS (18.9 U/g), respectively. These results show that the variations in PAL and PPO activity in the different organs of *P. chinense* seedlings depend on soil types and growth stages.

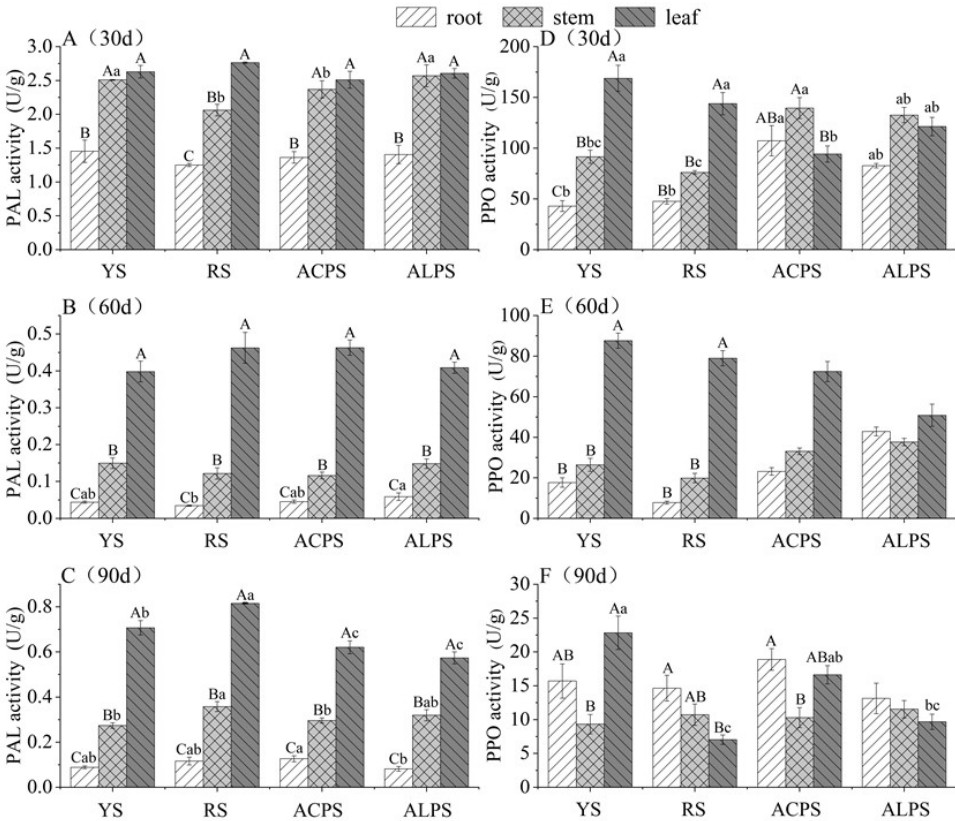

**Figure 5.** Effect of soil types on phenylalanine ammonialyase (PAL) and polyphenol oxidase (PPO) activities in *P. chinense* seedlings at three growth stages. (**A**), PAL activities in leaves, stems and roots of seedlings at 30 days in four soil types. (**B**), PAL activities in leaves, stems and roots of seedlings at 60 days in four soil types. (**C**), PAL activities in leaves, stems and roots of seedlings at 90 days in four soil types. (**D**), PPO activities in leaves, stems and roots of seedlings at 30 days in four soil types. (**E**), PPO activities in leaves, stems and roots of seedlings at 60 days in four soil types. (**F**), PPO activities in leaves, stems and roots of seedlings at 90 days in four soil types. YS, yellow soil. RS, red soil. ACPS, acidic purple soil. ALPS, alkaline purple soil. Data represent mean ± S.E., *n* = 3. Lowercase letters represent significant differences in different soil types, and capital letters represent significant differences in different organs, with a significance level of 0.05.

### 3.6. Principal Component Analysis (PCA)

As shown in Figure 6, the results from PCA show that a clear difference is visualized when the seedlings are cultured in four soil types at three growth stages. Principal component 1 (PC1) corresponds to biomass, osmoregulatory substance contents, and antioxidant enzyme activities in *P. chinense* seedlings. MDA, chlorophyll, total polyphenol, soluble protein, soluble sugar, SOD, POD, PPO, and PAL activities can explain most of the variation in the dataset, indicating that these variables contribute to a better understanding of the adaptation of *P. chinense* seedlings to the soil environment. In four soil types, the cumulative contribution percentages of PC1 and PC2 range from 74.44% to 81.97%, which can represent the original variables to a certain extent. In yellow soil (Figure 6A), the variance percentage of PC1 is 46.58%, which is mainly contributed to by the protein, soluble sugar, and total polyphenol contents, while the percentage of variation for PC2 is 32.01%, which is mainly determined by biomass and MDA content (Table 2). The percentage of variation for PC1 in RS (Figure 6B, Table 2) is 46.80%, which is mainly determined by protein, soluble sugar, and total polyphenol content, while the percentage of variation for PC2 is 35.17%, which is mainly determined by biomass and MDA content. As shown in Figure 6C, cultured in ACPS, the variance percentage of PC1 is 40.81%, which is mainly contributed to by the higher POD, PAL, and PPO activities, and the variance percentage of PC2 is 33.63%, with a main contribution of soluble sugars and total polyphenol content. When cultured in ALPS (Figure 6D), the variance percentage of PC1 is 43.84%, which is mainly contributed by the higher protein content, POD, PAL, and PPO activities, while the variance percentage of PC2 is 34.30% and the main contributions are total polyphenols content (Table 2). These results show that the main contributing components of PC1 varies in four soil types, and also significantly depends on growth stages and organs in *P. chinense* seedlings.

**Table 2.** Principal component load and contribution rate of physiological and biochemical indexes in *P. chinense* seedlings.

| Indexes | YS | | RS | | ACPS | | ALPS | |
|---|---|---|---|---|---|---|---|---|
| | PC1 | PC2 | PC1 | PC2 | PC1 | PC2 | PC1 | PC2 |
| Biomass | −0.01 | 0.56 | 0.01 | 0.54 | −0.34 | 0.35 | −0.29 | 0.42 |
| Soluble protein | 0.44 | 0.11 | 0.45 | −0.02 | 0.24 | 0.44 | 0.37 | 0.28 |
| Soluble sugar | 0.33 | 0.30 | 0.29 | 0.37 | −0.08 | 0.47 | −0.06 | 0.43 |
| Total polyphenols | 0.38 | 0.32 | 0.31 | 0.42 | 0.03 | 0.54 | 0.10 | 0.52 |
| Malondialdehyde (MDA) | −0.05 | 0.55 | −0.09 | 0.53 | −0.42 | 0.24 | −0.38 | 0.29 |
| Superoxide dismutase (SOD) | −0.18 | −0.19 | −0.22 | −0.16 | −0.04 | −0.23 | 0.02 | −0.37 |
| Peroxidase (POD) | 0.42 | −0.19 | 0.45 | −0.10 | 0.45 | 0.19 | 0.42 | 0.23 |
| Phenylalanine ammonialyase (PAL) | 0.38 | −0.26 | 0.42 | −0.19 | 0.46 | 0.14 | 0.46 | 0.09 |
| Polyphenol oxidase (PPO) | 0.44 | −0.21 | 0.42 | −0.23 | 0.47 | −0.05 | 0.48 | −0.05 |
| Eigenvalue | 4.19 | 2.88 | 4.21 | 3.17 | 3.67 | 3.03 | 3.95 | 3.09 |
| Percentage of variance (%) | 46.58 | 32.01 | 46.80 | 35.17 | 40.81 | 33.63 | 43.84 | 34.30 |
| Cumulative (%) | 78.58 | | 81.97 | | 74.44 | | 78.14 | |

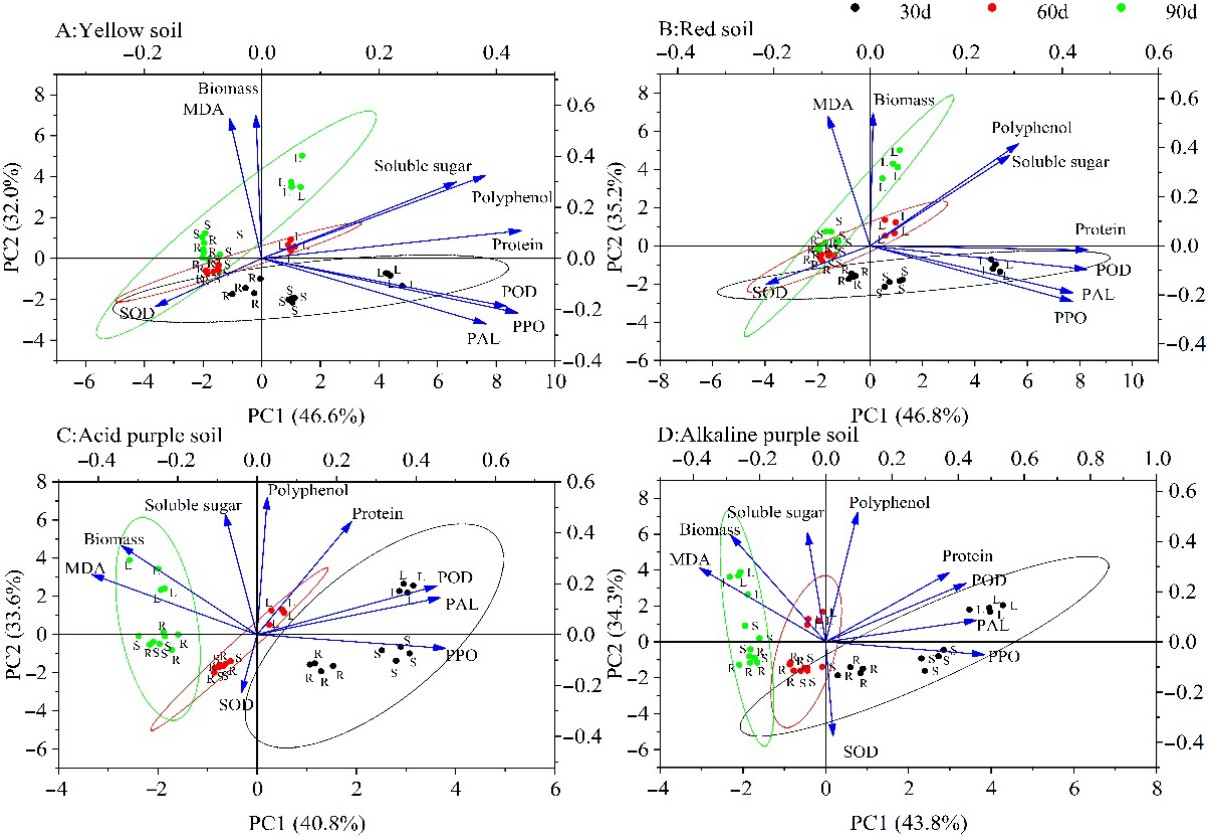

**Figure 6.** Principal component analysis (PCA) of biomass, physiological indexes of *P. chinense* seedlings in response to four soil types at three growth stages. (**A**) Yellow soil, (**B**) red soil, (**C**) acidic purple soil, and (**D**) alkaline purple soil. Loading of variables (MDA, malondialdehyde; SOD, superoxide dismutase; POD, peroxidase; PPO, polyphenol oxidase; PAL, phenylalanine ammonialyase; L, leaves; S, stems; R, roots) in PC1 and PC2 are shown by the direction and strength of the vector lines. Percent variation explained by each component is given next to the axis. Length of the variable vector shows a discriminative property of trait studied. Location of a trait in diagram closest to the intersection of 0 on the *X*-axis (PC1) and *Y*-axis (PC2) shows similarity.

## 4. Discussion

During the cultivation of seedlings, soils are one of the very key controlling factors, which provide a suitable environment for seedlings' growth and development [2,3]. Thus, how to select the effects of appropriate soil types on stimulation of the growth, development, and the underlying regulatory mechanisms in *P. chinense* remain to be further analyzed. This present study shows that *P. chinense* seedlings can grow in four soil types by regulating the growth parameters, biochemical contents, and antioxidant responses, and *P. chinense* seedlings cultivated in acidic purple soil exhibit better growth characteristics than those cultivated in other soils. Moreover, these characteristics have significantly positive correlations with soil types, growth stages, and tissues (Figure 6).

Plant height, base diameter, and biomass of seedlings are the basic indicators of plant growth, which can directly reflect the growth status of plants, and the plant–environment relationship [29]. The present study shows that the seedling height, basal diameter, and biomass of *P. chinense* seedlings show significant differences among four soil types and growth stages (Figure 1). This study also reveals that the ACPS represents the best comprehensive performances in both growth characteristics among the four tested soil types, followed by those of YS, RS, and ALPS. Shu et al. showed that *Jatropha curcas* seedlings grew well in red soil (RS) and yellow soil (YS), and plant height and base diameter were significantly higher than those in yellow–brown soil (YBS) and purple soil (PS) [30]. A

previous study pointed out that seedling height, basal diameter, and total biomass have significant differences when *Ricinus communis* seedlings were cultured in four soil types, and the maximum values were observed in alkaline purple soil [31]. These results show that soil types have a great influence on growth parameters and biomass allocation in many plants species, which may be partially due to the significant differences in pH, and the contents of organic matter, total N, total P, and total K, etc., among the tested four soil types (Table 1). This is possibly because of the large difference in soil nutrient requirements among plant species [2,3,21]. We determined that ACPS may be the suitable soil type for seedling cultivation of *P. chinense*, as supported by several lines of evidence as follows: (a) ACPS contained the most abundant organic matter, alkali-hydrolysable nitrogen; (b) ACPS displayed the best performance in several important agronomic trials. ACPS, and YS, are relatively fine textured with a high content of silt and clay, and, therefore, have a high water-holding capacity, and a high cation exchange capacity (Table 1), which usually provide a stable supply of higher nutrient content for the growth and development of plants. RS and ALPS have a lower content of soil colloids, a poor water-holding capacity, and usually a very low content of minerals and nutrient content (Table 1). Thus, plant growth in such soils is usually poor. The plants produced most roots in the sandy soil, probably in response to lower nutrient availability and looser soil texture. The other soils contained more organic matter and had higher CECs, diminishing the need for extensive root systems. Reports show that the types and texture of soil are closely related to the growth and development of medicinal plants and that loam soil is the relatively ideal type of soil for the cultivation of root/stem-type of medicinal plants [32]. Thus, different soil types with soil texture, water-holding capacity, and the changes in microbial and soil enzyme activities, etc., lead to a certain impact on the nutrient absorption and utilization rate of plant growth and development. Generally, the most favorable soil type for cultivating high-quality seedlings is ACPS. If other types of soil are used for seedling cultivating or large-scale planting, the appropriate addition of N and/or K should be performed for improving the soil texture, and mineral nutrient contents. This will help the growth and cultivation of *P. chinense* seedlings.

Photosynthetic pigments in plant leaves are the basis of photosynthesis, and the content of photosynthetic pigments indicates the strength of photosynthetic capacity. Within a certain range, the higher the content of photosynthetic pigments, the stronger the photosynthetic capacity of plants. It has been suggested that photosynthesis influences the production of carbohydrates and plant growth [33]. The reduction in chlorophyll content may create disturbance in photosynthesis and carbohydrate production, which may lead to less plant growth and oil yield, which is in complete agreement with observed reduction in the oil yield of peppermint due to disturbance in photosynthesis that occurred under stress conditions [34]. Sharma et al. [35] observed the maximum growth of cassia species in black soil followed by sandy and humus soils; the leaf pigments were also observed more in black soil than those in other soils. The chlorophyll content of wheat and soybean seedlings in different soil types was different [36,37]. In this study, there were significant differences in chlorophyll content among different soil types and cultivation periods. Maximum chlorophyll, total sugar, and total polyphenol content of the leaves was observed in plants grown in sandy clay soil (Figures 2 and 3). The present results also show the directly proportional relationship between plant growth and chlorophyll content (Figure 1). These results indicate that different soil types can affect the chlorophyll content of *P. chinense* seedlings, which is similar to the results of previous studies [21]. Moreover, as important osmotic regulatory substances and nutrients in plants, soluble protein, soluble sugar, and total polyphenol are important indicators of plant drought resistance, because the increase in and accumulation of their content can not only improve the water retention ability of cells but also play a role in protecting the living substances and biofilms of cells [38]. Leaf-soluble sugars produced by photosynthesis export from the source leaves into the phloem, and are used directly for plant growth. In general, the content of soluble protein and soluble sugar in leaves is significantly higher than that in stems and roots, because

soluble protein and soluble sugar are photosynthetic products, which need to be produced in leaves and then slowly transported to the lower stems and root organs along the sieve tube [39]. Earlier reports showed that the polyphenol content in plants grown in different types of soil varied with the growth stage and soil types [40,41]. This contrasting pattern between growth and leaf NSC in the four soil types implies that soil types associated with soil chemical and physical properties modify or even determine the availability of carbohydrates to plant growth.

Previous studies suggested that ROS play a critical role in the seed germination process and seedling establishment, especially at early seedling establishment stages [8,9]. However, excessive ROS can cause lipid peroxidation, membrane instability, protein degradation, and nucleotide injuries in plants, and consequently affect them, thus, making plants unable to grow normally, or even leading to plant death. MDA, known as a product of membrane lipid peroxidation, can indicate the extent of oxidative damage under stress conditions [42]. Increasing MDA content was observed in some plant species exposed to the stresses of heavy metals, salt, etc. [27,43,44]. For protection against oxidative damage, plant cells produce antioxidant enzymes, such as SOD, POD, PPO, and PAL, etc., which can effectively prevent the over-production of ROS, and further maintain normal growth and development. Thus, an induced increase in antioxidant enzyme activity has been considered an important mechanism in maintaining the balance of ROS metabolism against oxidative stress [10,41]. In the present study, the increase in MDA content and the variable SOD, POD, PPO, and PAL activities in the organs of *P. chinense* seedlings were observed during the early formation process, indicating an important role for these antioxidant enzymes in remitting oxidative stress induced by the four soil types. This is similar to the research results that show that the antioxidant enzymes of *Dalbergia odorifera* seedlings are different when cultured in different substances [45]. This may be related to the different physical and chemical properties of the four soil types, which profoundly influenced the seedlings' growth, and then adjusted the activities of various antioxidant defensive molecules to maintain normal growth and development. Previous studies have reported increases and/or decreases in SOD, POD, PPO, and PAL activities at early seedling establishment stages in some plants in response to soil types and growth stages, for example, *Jatropha curcas*, *Brassica napus*, *Spinacia oleracea*, and *Quercus ilex*, etc. [28,46–48]. Such findings suggest that this antioxidant enzyme may be up-regulated or inhibited in the early seedling establishment of *P. chinense* seedlings.

PCA analysis shows that there are significant variables of tested indexes in *P. chinense* seedlings in response to soil types, growth stages, and organs, and the cumulative contribution percentages of PC1 and PC2 range from 74.44% to 81.97%. Notably, the coordinates of the tested indexes in leaves are closer to the side of the PC1 axis where the variables are concentrated compared with those of in the stems and roots, suggesting that these osmoregulatory compounds (total polyphenol, soluble sugar, and protein) and antioxidant enzymes (POD, PAL, and PPO) in the leaves are sensitive to soil types and growth stages (Figure 6). These variable parameters, for example, total polyphenol, soluble protein, soluble sugar, POD, PAL, PPO, etc., make for good indexes for physiological responses at early seedling establishment stages to different soil chemical and physical properties. Moreover, these parameters also show significant differences in *P. chinense* seedlings cultured in acidic purple soil and alkaline purple soil compared to those cultured in yellow soil and red soil (Figure 6), indicating that these seedlings experience a greater stress in purple soil than those of in red soil and yellow soil. This may be related to the external pH of ACPS (4.343) and ALPS (8.733), which is known to influence the rate of cation uptake in plants. In the purple soil, the seedling growth is slightly lower than on the other soils, maybe because of the low pH, which influences the availability of many ions and minerals in the soil, is more mobile under acidic conditions, and can have phytotoxic effects [49]. At low external pH, the net uptake of monovalent cations is generally lower than at neutral pH, and at pH < 4 a net loss of cations may occur. This effect of pH is explained by competition between $H^+$ and the monovalent cations for the transporters in the plasma membrane and a lower electrochemical gradient across the plasma membrane at low external pH [50]. In

southwest China, *P. chinense* can grow well in acidic purple soil at different growth stages, and the higher phellodendrine content of its bark is significantly higher than those of other soil types [18,19]. This may be due to the fact that *P. chinense* might adjust themselves to tolerate external pH via an effective physiological defensive mechanism, and further maintain the normal growth and development as well as secondary metabolites synthesis. The present results support this hypothesis and provide a better understanding of the adaptive mechanism of *P. chinense* plants exposed to various soil types.

## 5. Conclusions

In summary, the current study provides a promising approach to select optimal soil types for cultivating high-quality seedlings. Findings allow us to conclude that *P. chinense* seedlings can grow in four soil types, and ACPS is beneficial for the artificial cultivation of *P. chinense* seedlings at different growth stages, relative to those of RS, ALPS, and YS. Moreover, significant varying changes in growth parameters and antioxidant responses of *P. chinense* seedlings are recorded, which are sensitive to soil types, growth stages, and different tissues. The changes in activities of SOD, POD, PAL, and PPO suggest that *P. chinense* seedlings are trying to adapt to different soil conditions, and exhibit satisfactory seedling establishment. This information provides not only effective ways for farmers/gardeners to select soil for conducting artificial cultivation of *P. chinense* seedlings, but also contributes to a better understanding of the adaptation of *P. chinense* seedlings at early growth stages to the soil environment. However, the relationship between soil environment, seedling growth, and development is very complex. Thus, future efforts should work towards the interactions among soil types and their interactions with growth stages, and long-term detailed field research across a broad range of soil types is needed for large-scale cultivation and wide planting of *P. chinense* plants.

**Author Contributions:** Conceptualization, G.-X.C. and S.G.; methodology, Y.Y. and Y.-Y.H.; software, validation, and formal analysis, Y.Y., W.-Z.Q. and Y.-J.W.; investigation, Y.Y. and Y.-Y.H., W.-Z.Q. and Y.-J.W.; writing—original draft preparation, Y.Y., G.-X.C. and S.G.; writing—review and editing, Y.Y. and Y.-Y.H., W.-Z.Q., Y.-J.W., H.-Y.R., S.G. and G.-X.C.; visualization and supervision, S.G. and G.-X.C. All authors have read and agreed to the published version of the manuscript.

**Funding:** This research was funded by the National standardization project of traditional Chinese Medicine (grant no. 2YB221-Y-SC-41).

**Data Availability Statement:** The data presented in this study are available on request from the corresponding author.

**Acknowledgments:** We are grateful to all of the group members and workers for their assistance in the field experiment.

**Conflicts of Interest:** The authors declare no conflict of interest.

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
