# Peer review of "Early Growth Characterization and Antioxidant Responses of Phellodendron chinense Seedling in Response to Four Soil Types at Three Growth Stages"

_forests, doi:10.3390/f14091746_

Round 1

Reviewer 1 Report

The manuscript entitled "Early Growth Characterization and Antioxidant Responses of Phellodendron chinense Seedling in Response to Four Soil Types at Three Growth Stages" is an interesting study dealing with the growth responses and biochemical attributes of P. chinense plant seedlings under four soil types. The work is very comprehensive and results are interesting. Authors have done a 90 days study and measured different growth and antioxidant responses at three growth stages i.e. at 30, 60 and 90 days. Significant differences in different parameters were observed with the soil types and growth stages. Overall, the findings and conclusion of the study are sound and may enrich the content of the journal. The flow of the content is fine and language is OK. However, there are a few concern which need to be addressed appropriately. Particularly, the major limitation of the study is the single species. Nowadays multispecies studies are being recommended for better understanding of the ecological phenomenon. Secondly, the variations in soil properties in addition to the climatic variability i.e. precipitation and temperature conditions of different sites might be the primary cause of the variations observed in the study. However, the information on these aspects are given less emphasis. The statistical analysis part can be elaborated. The results are too elaborate and can be reduced to make it clearer. Discussion can be improved. There are a few specific suggestions in the attached file which can be incorporated in the revised manuscript. There are a number of typological errors in the manuscript which need to be curated.

There are a number of typos in the manuscript and a few sentences need to be rephrased for more clarity.

Author Response

  1. Line 31, 114, 130:

Note: except soil, all other climatic parameters viz. precipitation and temperature, etc. were similar at all the studied sites? As these parameters are the major determinants of the seedlings growth.

so the temperature, precipitation and other climatic conditions were similar for all the study sites?

so the edaphic and climatic conditions were similar for all the studied sites?

Reply: Ok. Thanks for these questions, the experiment site was located at Sichuan Agricultural University, Chengdu, China at latitude 30°38' N, longitude 103°45' E. Moreover, seedling culture were carried out in a greenhouse under controlled conditions. Thus, precipitation and other climatic conditions have been excluded, and growth environment were similar for seedlings growth and developments.

  1. Keywords:

arrange alphabetically replace the words already present in title with other relevant words!

Reply: OK. We agree. We have revised this question in the revised manuscript as suggested.

  1. line 37, 42: reference neeed ?

Reply: OK. Thanks. For line 37, we have added a reference to support “The seedling stage is the most critical life-history stage for the normal development of plant systems and processes”.

  1. Line 53-55: “These reports indicated that soil types are directly related to growth, development physiological and biochemical parameters as well as the seedling quality of plants [6].” Note: In addition, inter- and intra-competition with seedlings and grasses play a major role in determining the seedlings performance.

Reply: Ok. We agree. The sentence have been added in the revised manuscript as suggested.

  1. 2. Soil Sampling and Chemical Analysis:

Soil treatment methodology needs to be rewritten to make it clearer.

Reply: Ok. We agree. We have corrected some descriptions of soil treatment methodology in the revised manuscript as suggested.

  1. 3. Seedling Culture and Growth Parameters Measurement

Note: The only limitation of this work is the number of species... only one species....  Nowadays multispecies studies are more preferable.

Reply: OK. We agree. Ok. We agree. Thank you very much for pointing out this question. Granted, only one species was a limitation. Of course, the comparison of multispecies cultured in different soil types and/or other conditions will help to understand the responses of growth, physiological and biochemical among these plant species. In the present study, we only used P. chinense seedlings as an example, which mainly provide a promising way to selecting optimal soil types for cultivating high-quality seedlings. Future studies will focus on the comparison of multispecies cultured in different soil types, which will help to better understand and assess the effects of different soil types on the early growth characterization and antioxidant responses in multi plant species.

  1. Table 1:

Note: so there is a significant differences between different parameters! these variations might be the possible cause of variation in growth and biochemical parameters of seedlings!

Reply: Ok. We agree. We have corrected some descriptions in the revised manuscript as suggested.

  1. Table 1, Moisture Content need a unit

Reply: OK. Thanks. The unit of moisture content have been added in the revised manuscript as suggested.

  1. 9. Determination of Phenylalanine Ammonia-lyase (PAL) Activity

Note: I think the methods are repetitive, i.e. already established methods and therefore can be shortened by giving reference to the original source.

It may also reduce the similarity content of the manuscript, if any?

Reply: Ok. We agree. We have shorten these description in the revised manuscript as suggested.

  1. 11. Statistical Analysis

PCA section can be elaborated.

normality of data was tested? which type of correlation was applied? which type of regression analysis was done?

Reply: OK. Thanks. The detail description about PCA in the Chapter of Statistical Analysis has been added in the revised manuscript as suggested. PCA is a commonly used analysis method for reducing data dimensionality (multiple random variables). The main principle is that the measured data was firstly centered, and the covariance matrix was calculated, further obtain the eigenvalues and eigenvectors. Thus, it reconstruct a new coordinate system based on the size of the eigenvalues to achieve de-dimension and standardization processing. In most analysis, PCA mainly starts from the perspective of transforming coordinate systems and matrix solving, and does not include normal analysis, correlation testing, and regression analysis of data. Thanks for your this question.

  1. Reduce results section at least by 30%.

Reply: Ok. Thanks. We have shorten the text and description of results section in the revised manuscript as suggested.

  1. 1 to 3.3 can be merged and presented in one for more clarity. highlight only important observations having novelty!

Reply: Ok. Thanks. The sentence has been rewrite from 3.1 to 3.3 in the revised manuscript as suggested.

  1. for figures: the font size of x- and y axis labels can be increased for more clarity!

Reply: Ok. Thanks. The font size of x- and y axis labels has been adjusted in the revised manuscript as suggested.

  1. Reduce PCA section at least by 20-25%!

Reply: Ok. Thanks. We have shorten the text and description of PCA section in the revised manuscript as suggested.

  1. Line 448-449:“The PCA showed a clear differentiation among soil types, growth stages and tissues in P. chinense seedlings. As shown in Figure 6A, cultured in yellow soil, the variation (46.6%) is explained by PC1, and PC2 contributes 32.0% in total this biplot explained 78.6% of the variation. ”

rephrase the sentence for more clarity!

Reply: Ok. Thanks. The sentence has been revised in the revised manuscript as suggested.

  1. rewrite PCA section with more focus on the important relations having significance in explaining the key observations of the study!

Reply: Ok. Thanks. The sentence has been rewrite in the revised manuscript as suggested.

  1. Table 2 :elaborate the abbreviations or mention table/figure numbers where they are elaborated!

Reply: Ok. Thanks. The abbreviations has been noted in the revised manuscript as suggested.

  1. Line 574-576:“These results indicated that different soil types could affect the chlorophyll content of chinense seedlings, which was similar to the results of previous studies.” need to cite a few studies here!

Reply: Ok. Thanks. The reference has been added in the revised manuscript as suggested.

  1. There are many errors in the manuscript such as incorrect capitalisation, incorrect singular and plural numbers, no spaces between words, etc., which need to be corrected

Reply: Ok. We agree. We have revised the errors or miswriting as suggested.

Comments on the Quality of English Language

There are a number of typos in the manuscript and a few sentences need to be rephrased for more clarity.

Reply: Ok. Thanks. The text of the whole paper has been revised, and the language was improved by a native speaker as suggested.

Reviewer 2 Report

Dear Authors,

Based on following two things,

1. poor language

2. lack of novelty

Needs Major Revision.

It is not feasible to mention all the grammatical mistakes, however, sentences/paragraphs are mentioned as follows for major revision. 

1. Rephrase the sentence (Line No. 36-37).

2. Rephrase the sentence (Line No. 44-51).

3. Rephrase the sentence (Line No. 53-55).

4. Rephrase the sentence (Line No. 60-64).

5. Correct “earlyreports” as “early reports” (Line No. 64).

6. Sentence need rephrasing (Line No. 66-68).

7. Correct “change in non-enzymatic” as “changes in non-enzymatic” (Line No. 69).

8. Add a “comma” after although (Line No. 82).

9. Replace “Soil type” as “soil type” (Line No. 88).

10. Write full form of TN at first use (Line No. 128).

11. Add reference of ignition method or give full detail of ignition method (Line 130).

12. Rephrase the sentence (Line No. 141-144).

13. Write sentences in past (Line No. 160-162).

14. Write full form of POD, SOD and PPO at first use (Line No. 170).

15. Rephrase the sentence (Line No. 174-175).

16. Rephrase the sentence (Line No. 182-183).

17. Write full form of NBT at first use (Line No. 191).

18. Correct “12 000 rpm” as “12,000 rpm” (Line No. 169, 184, 206).

19. Correct “100μl” as “100 μl” (Line No. 210).

20. Rephrase the sentence (Line No. 214-220).

21. In results section, “cultured term” is frequently used, although it makes no appropriate sense, need re-consideration.

22. All the results have been written in copy paste format, need re-consideration.

23. In results section, the effect of different soil types on individual growth and biochemical parameters in particular plant organ (for instance root) is non-significant. It’s understood that root biomass will be different from shoot but the differential effect of soil types on root biomass is non-significant which is the major objective of this article. Same finding has been observed for other measured parameters, need re-consideration.

24. Rephrase the sentence (Line No. 505-512).

25. Rephrase the sentence (Line No. 520-524).

26. Rephrase the sentence (Line No. 534-543).

27. Rest paragraphs also need re-consideration and compaction.

Quality of English language is very low. Sentences has been made with 

Author Response

  1. Rephrase the sentence (Line No. 36-37).

“while the demands of different seedlings have different demands for suitable soil conditions due to different needs of nutrients, water, suitable pH, etc.”

Reply: Ok. We agree and the text was changed in the revised manuscript as suggested.

  1. Rephrase the sentence (Line No. 44-51).

“rameters on different soil types were significantly different, and plants growing in sandy 45 clay had the fastest growth and the highest content of essential oil yield inMentha arvensis 46 [1]. When comparing Artemisia Annua plants under clay loamy and sand loamy soils, 47 higher vegetative growth characters were obtained under clay loamy soil [3]. The height 48 and diameter growth of Cyclobalanopsis glaucoides seedlings on black lime soil were sig- 49 nificantly higher than those on red lime soil. Some studies have shown that higher soil 50 moisture and nutrient availability are more important in the early than those in the later 51 stages of seedling growth [4,5]. ”

Reply: Ok. We agree and the text was changed in the revised manuscript as suggested.

  1. Rephrase the sentence (Line No. 53-55).

“These reports indicated that soil types are directly related to growth, development physiological and biochemical parameters as well as the seedling quality of plants [6]. ”

Reply: Ok. We agree and the text was changed as suggested.

  1. Rephrase the sentence (Line No. 60-64).

“In order to balance the production of excess ROS between ROS production and detoxification, plants have therefore induced non-enzymatic (soluble sugars, soluble protein, polyphenol, and amino acid, etc.) and enzymatic (superoxide dismutase, catalase, peroxidase, phenylalanine ammonia-lyase, polyphenol oxidase) systems enabling them to counteract these oxidative damages [8].”

Reply: Ok. We agree and the text was changed as suggested.

  1. Correct “earlyreports” as “early reports” (Line No. 64).

Reply: Ok. We agree. We have revised the miswriting as suggested.

  1. Sentence need rephrasing (Line No. 66-68).

“Previous studies have mentioned that different soils (salinity, water content, element content, etc.) affect antioxidant enzyme activities in plants [10-12], and was directly related to seed germination and early seedling survival [13,14]. ”

Reply: Ok. We agree and the text was changed as suggested.

  1. Correct “change in non-enzymatic” as “changes in non-enzymatic” (Line No. 69).

Reply: Ok. We agree. We have revised the miswriting as suggested.

  1. Add a “comma” after although (Line No. 82).

Reply: Ok. We agree. We have revised the miswriting as suggested.

  1. Replace “Soil type” as “soil type” (Line No. 88).

Reply: Ok. We agree. We have revised the miswriting as suggested.

  1. Write full form of TN at first use (Line No. 128).

Reply: Ok. We agree. We have revised this question as suggested.

  1. Add reference of ignition method or give full detail of ignition method (Line 130).

“The soil moisture and organic matter content were determined by loss on ignition method.”

Reply: Ok. Thanks. The reference has been added in the revised as suggested.

  1. Rephrase the sentence (Line No. 141-144).

“After germination for 7 days, until the emerging seedlings presented their fully extended cotyledonal leaves (BBCH10 stage) approximately 30 days after sowing seeds, the seedlings with poor growth were removed, and only one seedling was left in each pot.”

Reply:Ok. We agree. We have rewrite this sentence in the revised manuscript as suggested.

  1. Write sentences in past (Line No. 160-162).

Reply: Ok. We agree. We have rewrite this sentence as suggested.

  1. Write full form of POD, SOD and PPO at first use (Line No. 170).

Reply: Ok. We agree. The full names of POD, SOD and PPO have been added as suggested.

  1. Rephrase the sentence (Line No. 174-175).

“The reaction solutions contained 0.4 ml extraction and 0.4ml of Folin-Ciocalteau phenol reagent, and stand for 5 min. ”

Reply: Ok. We agree. We have rewrite this sentence as suggested.

  1. Rephrase the sentence (Line No. 182-183).

“To a 1.0 ml aliquot of the supernatant, 4.0 ml of 0.5% TBA in 5% TCA was added, and these mixtures were heated at 100°C for 30 min and then cooled in an ice bath. ”

Reply: Ok. We agree. We have rewrite this sentence as suggested.

  1. Write full form of NBT at first use (Line No. 191).

Reply: Ok. We agree. The full name of NBT has been added as suggested.

  1. Correct “12 000 rpm” as “12,000 rpm” (Line No. 169, 184, 206).

Reply: Ok. We agree. We have revised this questions as suggested.

  1. Correct “100μl” as “100 μl” (Line No. 210).

Reply: Ok. We agree. We have revised this questions as suggested.

  1. Rephrase the sentence (Line No. 214-220).

“2.10. Determination of Polyphenol Oxidase (PPO) Activity

Enzyme extracts are prepared in the same way as SP extracts. PPO activity was assayed by the method of He et al. [23]. PPO activity was measured by incubating 0.5 ml of enzyme extract to 2.5ml buffered substrate (100 mM sodium phosphate, pH 6.4 and 50 mM Catechol), and then monitoring the change of absorbance at 398 nm, one unit of PPO activity was defined as the amount of enzyme causing 1 absorbance increase per min, and data was expressed as units per gram fresh weight (U/g FW).”

Reply: Ok. Thanks. We have rewrite this sentence as suggested.

  1. In results section, “cultured term” is frequently used, although it makes no appropriate sense, need re-consideration.

Reply: Ok. Thanks. We have corrected some description as suggested.

  1. All the results have been written in copy paste format, need re-consideration.

Reply: Ok. We agree. We have rewrite the description of results in the revised manuscript as suggested.

  1. In results section, the effect of different soil types on individual growth and biochemical parameters in particular plant organ (for instance root) is non-significant. It’s understood that root biomass will be different from shoot but the differential effect of soil types on root biomass is non-significant which is the major objective of this article. Same finding has been observed for other measured parameters, need re-consideration.

Reply: Ok. Thanks. We have corrected some description as suggested.

  1. Rephrase the sentence (Line No. 505-512).

“4. Discussion

As a traditional Chinese medicinal woody plant, the increasing market demand for P. chinense is contradictory to the limited supply source due to the longer harvest cycle from 8 to 12 years[15,18]. One of the effective ways is artificial cultivation on large-scale, which depends on the high-quality seedlings of P. chinense. During the cultivation of seedlings, soils are one of very key controlling factors, which provide suitable environment for seedlings' growth and development [1,2]. Thus, how to select the effects of appropriate soil types on stimulation of the growth, development, and the underlying regulatory mechanisms in P. chinense remain to be further analyzed. ”

Reply: Ok. Thanks. Some descriptions have been revised in the revised manuscript as suggested.

  1. Rephrase the sentence (Line No. 520-524).

“Present study showed that seedling height, basal diameter and biomass of P. chinense seedlings were significant differences among four soil types, and are related to growth stages (Figure 1). This study also revealed that the acidic purple soil represents the best comprehensive performances in both growth characteristics among four tested soil types, followed by those of yellow soil, red soil, and alkaline purple soil in the order from high to low. ”

Reply: Ok. Thanks. Some descriptions have been revised in the revised manuscript as suggested.

  1. Rephrase the sentence (Line No. 534-543).

“This is possible because of the large difference in soil nutrient requirements among plant species. We determined that acidic purple soil may be the suitable soil type for seedling cultivation of P. chinense, as supported by several lines of evidence as follows: (a) acidic purple soil contained the most abundant organic matter, alkali-hydrolysable nitrogen; (b) acidic purple soil displayed the best performances in several important agronomic trials. Acidic purple soil, and yellow soil, are relatively fine textured with a high content of silt and clay and therefore have a high water holding capacity, a high cation exchange capacity (Table 1), ……”

Reply: Ok. Thanks. We have rewrite the description of this sentence in the revised manuscript as suggested.

  1. Rest paragraphs also need re-consideration and compaction.

Reply: Ok. Thanks. We have rewrite the description of these paragraphs in the revised manuscript as suggested.

Comments on the Quality of English Language

Quality of English language is very low. Sentences has been made with 

Reply: Ok. Thanks. The text of the whole paper has been revised, and the language was improved by a native speaker as suggested.

Round 2

Reviewer 1 Report

The manuscript has been revised substantially. Authors' effort in revising the manuscript must be appreciated.  However, there are a few minor concerns related to some sentences and reference citations in the text which need to be curated. See the attached file for more details.

There are a few sentences which need clarification or can be rephrased for better clarity.

Author Response

Comments 1 Lowercase the first letter "Chinense" in the title of the article.

Reply 1: OK, We agree. We have revised this question as suggested.

Comments 2 Keyword: include 2-3 additional relevant keywords!

Reply 2: OK. Thanks. We have added 2-3 additional relevant keywords as suggested.

Comments 3 Line 51-53: check reference citations and if it is said in the cited article or see other articles cited earlier (e.g. citation 1)!

Reply 3: OK, We agree. We revised the relevant references according to this suggestion.

Comments 4 Line 60:Insert a space between "peroxidase " and "(POD)".

Reply 4: Ok. We agree. We have inserted a space as suggested.

Comments 5 Line 122: “property indexes” can be rewritten AS “properties”.

Reply 5: OK, thanks. We have revised this question as suggested.

Comments 6 Line 237: elaborate the abbreviations in the section/sub-section names throughout the manuscript!

Reply 6: Ok. We agree. We have made the relevant changes as suggested. Other similar issues in the article have made relevant changes

Comments 7 Line 365-366: something is missing or sentence seems to be incomplete!

Reply 7: Ok. We agree. We have made the relevant changes as suggested.

Comments 8 Line 366-369: rephrase the sentence for more clarity!

Reply 8: Ok. We agree. We have made the relevant changes as suggested.

Comments 9 Line 404-408: rephrase the sentence for clarity!

Reply 9: Ok. Thanks. The sentence has been rewrite as suggested.

Comments 10 Line 423: change “ clay and therefore” into “ clay, and therefore,”

Reply 10: OK, thanks. We have revised this question as suggested.

Comments 11 Line 531: “ to selecting” need to modify.

Reply 11: OK, thanks. We have revised this question as suggested.

Comments 12 check reference formatting throughout the manuscript and the references section. there are some discrepancies in the presentation of citations in the list.

For example, 1. seems incomplete; 3. year and volume, and 6. seems not presented correctly.  Also cross check others.

  1. Bhadouria, R.; Singh, R.; Srivastava, P.; Raghubanshi, A.S. Understanding the ecology of tree-seedling growth in dry tropical environment: a management perspective. Energ. Ecol. Environ., 2016, 1, 296–309.
  2. Heimler, D.; Romani, A.; Ieri, F. Plant polyphenol content, soil fertilization and agricultural management: a review. Eur. Food Res. Technol., 2017, 243, 1107–1115.
  3. Clause, J.; Barot, S; Forey, E. Effects of cast properties and passage through the earthworm gut on seed germination and seedling growth. Appl. Soil Ecol., 2015, 96,108-113.

Reply 12: OK, thanks. The format of these references has been checked and corrected as suggested.

Comments on the Quality of English Language

There are a few sentences which need clarification or can be rephrased for better clarity.

Reply: Ok. Thanks. We greatly appreciate your help for concerning improvement to this paper. We are so sorry for our careless and bringing you trouble. The sentences of the manuscript have been checked and corrected, and the language was improved as suggested.

Reviewer 2 Report

I have gone through the revised version, suggestions have been incorporated.

 Minor editing of English language required

Author Response

Comments and Suggestions for Authors

I have gone through the revised version, suggestions have been incorporated.

Comments on the Quality of English Language

 Minor editing of English language required

Reply: Ok. Thanks. We greatly appreciate your help for concerning improvement to this paper. We are so sorry for our careless and bringing you trouble. The sentences of the manuscript have been checked and corrected, and the language was improved as suggested.